# Skywork-Math: Data Scaling Laws for Mathematical Reasoning in LLMs — The Story Goes On

**Liang Zeng**
Skywork AI
liang.zeng@kunlun-inc.com

**Liangjun Zhong**
HKUST
lzhongah@connect.ust.hk

## Abstract

In this paper, we investigate the underlying factors that potentially enhance the mathematical reasoning capabilities of large language models (LLMs). We argue that the data scaling law for math reasoning capabilities in modern LLMs is far from being saturated, highlighting how the model's quality improves with increases in data quantity. To support this claim, we introduce the Skywork-Math model series, supervised fine-tuned (SFT) on common 7B LLMs using our proposed 2.5M-instance Skywork-MathQA dataset. Skywork-Math 7B has achieved impressive accuracies of 51.2% on the competition-level MATH benchmark and 83.9% on the GSM8K benchmark using only SFT data, outperforming an early version of GPT-4 on MATH. The superior performance of Skywork-Math models contributes to our novel two-stage data synthesis and model SFT pipelines, which include three different augmentation methods and a diverse seed problem set, ensuring both the quantity and quality of Skywork-MathQA dataset across varying difficulty levels. Most importantly, we provide several practical takeaways to enhance math reasoning abilities in LLMs for both research and industry applications.

## 1 Introduction

*More is different.*

———-Philip W. Anderson, 1972

Reasoning ability is a hallmark of human intelligence [14, 11, 27]. Although Large Language Models (LLMs) have recently demonstrated significant capabilities in various tasks such as conversation [1, 3, 18] and summarization [28, 30, 21, 2], they often struggle with complex reasoning tasks [11, 17, 29]. One particularly challenging area is mathematical reasoning [13, 9, 32, 4, 12], which requires the ability to solve mathematical problems and derive logical conclusions in a step by step manner [27, 20, 23, 31, 24].

Two prevailing beliefs guide researchers and practitioners in enhancing mathematical reasoning abilities of LLMs. The first belief posits that complex reasoning abilities, especially mathematical reasoning, are emergent abilities that exist in large language models but not in small models [27, 26]. Typically, models with more than 30 billion parameters exhibit the strong mathematical reasoning ability [7]. The second belief is the seminal "superficial alignment" hypothesis [33], which asserts that *"A model's knowledge and capabilities are learnt almost entirely during pre-training, while alignment teaches it which sub-distribution of formats should be used when interacting with users."*. According to this hypothesis, the alignment process, primarily through supervised fine-tuning (SFT), does not inject new knowledge or improve inherent abilities but rather adjusts the output response format. This implies that the strong mathematical reasoning ability may not be significantly improved by a large amount of synthetic SFT data.

38th Conference on Neural Information Processing Systems (NeurIPS 2024).

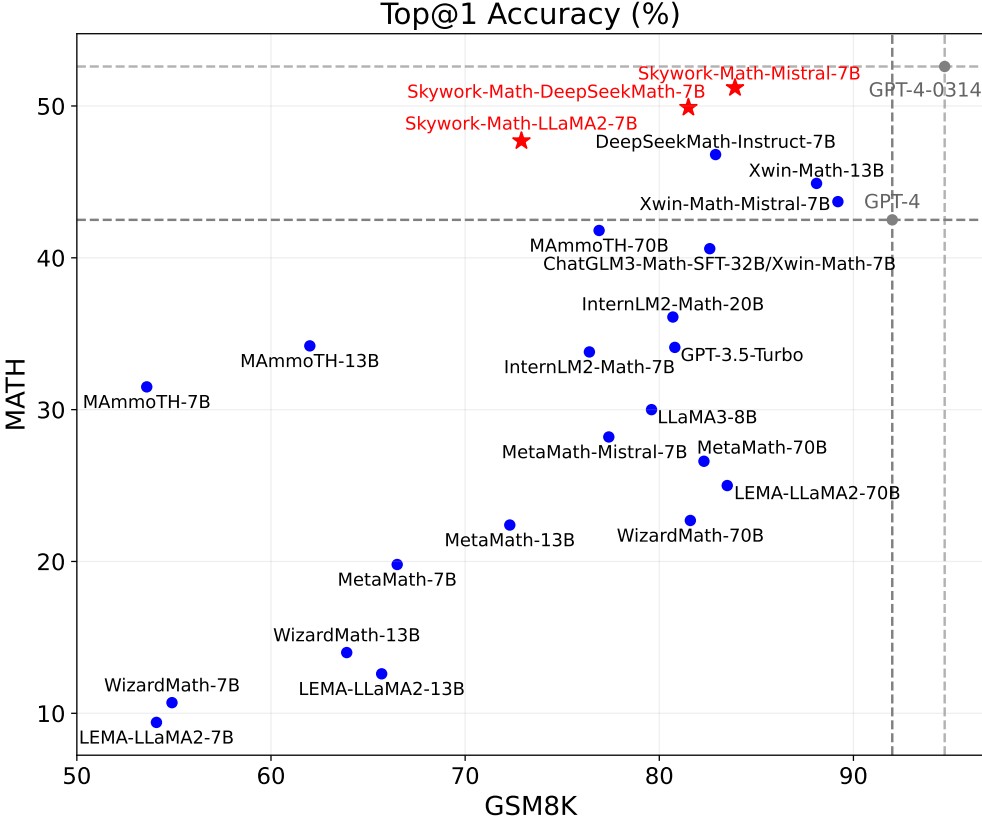

Figure 1: Top1 accuracy on GSM8K [9] and MATH [13] using only SFT techniques, without using external toolkits and voting techniques. Following MetaMath [31], we employ a zero-shot chain-of-thought evaluation framework. Skywork-Math models achieve state-of-the-art accuracy among models smaller than 10B parameters using only synthetic SFT data and surpass an early version of GPT-4 on MATH.

In this paper, we re-examine these two common beliefs mentioned above regarding mathematical reasoning abilities of LLMs. For the first belief, we introduce the Skywork-Math model series, which are supervised fine-tuned (SFT) on common 7B pre-trained LLM models without employing other complex alignment techniques such as RLHF [6, 8] and DPO [19]. Skywork-Math 7B models have achieved impressive accuracies of 51.2% on the competition-level MATH [13] benchmark and 83.9% on the GSM8K [9] benchmark, notably outperforming an early version of GPT-4 on MATH. Our empirical findings, consistent with the conclusions in [16], suggest that strong mathematical reasoning ability can indeed exist in common 7B language models. Moreover, scaling up synthetic SFT data can further enhance the mathematical reasoning ability of Skywork-Math 7B models.

For the second belief, we propose Skywork-MathQA high-quality SFT dataset containing 2.5 million instances, which is much larger than open-sourced dataset of its kind to date, such as Meta-MathQA [31] containing 395K samples. We empirically observe that the scaling law curve on the SFT alignment for mathematical reasoning in modern LLMs is far from being saturated (ref. Figure 3). We have carefully scaled the Skywork-MathQA SFT dataset with diverse and high-quality samples specifically within the mathematical domain to enhance the model's capability in understanding and solving mathematical problems.

Due to the scarcity of high-quality and challenging mathematical data, various pipelines and prompts have been employed to generate synthetic mathematical data [31, 23, 16, 24, 27, 25]. To address this deficiency, we employ GPT-4 to generate a substantial amount of synthetic data through a novel two-stage data synthesis pipeline, in conjunction with the corresponding model SFT process. In stage 1, our objective is to obtain normal synthetic problems to enhance the models' general comprehension of mathematical problems. To maintain the diversity in data selection process, we utilize the core-set approach [22] on enlarged seed problems. However, as the data volume increases, we empirically

observe that the relationship between performance and data quantity begins to plateau. Accordingly, in stage 2, we diversify the dataset further by introducing a proportion of augmented hard problems, thereby exposing the model to more challenging mathematical questions. Without continual pre-training on a large-scale math corpus [23, 5], Skywork-Math models achieve impressive performance with just supervised fine-tuning on common pre-trained LLMs containing only 7B parameters.

**Highlighted Takeaways**

- The potential for math reasoning capabilities in modern LLMs is far from exhausted. The quality of LLMs can significantly improve with increases in data quantity (ref. Figure 3). Skywork-Math 7B models already demonstrate strong mathematical reasoning abilities by SFTing on common 7B pre-trained LLM models.

- The learning process for accessing the math reasoning ability involves multiple stages. Training LLM models in a meaningful order, from the easy problems to the hard ones, can provide performance improvements.

- When scaling the synthetic SFT dataset, increasing the diversity of seed problems and augmentation methods can improve the math reasoning performance of LLMs.

- Selecting influential data with high-quality from a large dataset is non-trivial [10]. Our empirical results indicate that some straightforward methods to select the so-called "high-quality" data may not increase (and can even hurt) LLMs' performance compared to randomly selecting data. The selection process involves multiple constraints, and the "high-quality" data could significantly decrease the difficulty level of problems, thus negatively impacting the performance of LLMs.

- The LLM models have strong knowledge transfer capabilities for mathematical reasoning across bilingual benchmarks (i.e., English and Chinese). We hypothesize that this can be attributed to the inherent nature of symbols and numbers in math problems, which retain their intrinsic meaning regardless of the language used.

- Although Skywork-Math 7B models have achieved considerable improvement in robustness tests compared to other open-source LLM models, they remain sensitive to the distractors in math word problems compared with proprietary GPT-4 models.

- Sparse MOE models cannot clearly exceed the performance upper bound of their dense counterparts through SFT alignment in the context of math reasoning.

- Two subtle but crucial practical implementation techniques—preventing data leakage and considering the influence of model maximum length—significantly impact the final performance of LLM models.

Most importantly, we provide valuable insights and practical takeaways to enhance the mathematical reasoning ability in LLMs, benefiting both research and industry communities 2.

## 2   Method

In this section, we present the methodology of Skywork-Math 7B models, as illustrated in Figure 2. Skywork-Math models aim to enhance math reasoning abilities during the model alignment process, particularly in the SFT stage, using common and publicly available 7B pre-trained models. We employ a two-stage SFT approach, in conjunction with two data synthesis pipelines to produce high-quality data. In stage 1, we feed base pre-trained models with our generated normal synthetic problems (2.1M instances) to produce an intermediate model. In stage 2, to mitigate the diminishing returns in LLMs' performance as the quantity of data increases, we generate hard synthetic problems (0.4M instances) and develop our Skywork-Math models. To ensure the quality of data, we primarily utilize GPT-4-1106-preview [1] to generate 2.5M-instance synthetic Skywork-MathQA dataset. Due to space constraints, detailed methods and experimental results can be found in the appendix.

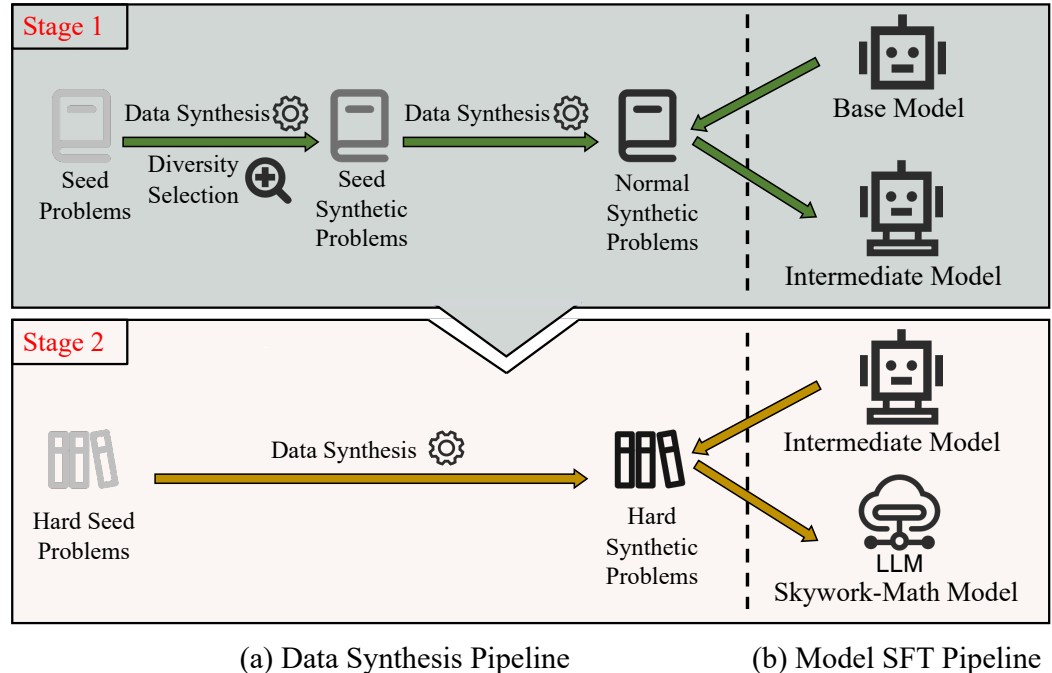

(a) Data Synthesis Pipeline          (b) Model SFT Pipeline

Figure 2: Overview of our proposed two-stage method. (a) The data synthesis pipeline of the Skywork-MathQA dataset. (b) The model SFT pipeline of the Skywork-Math model series.

# 3 Data Scaling Laws in SFT on Mathematical Reasoning

In Figure 3, we illustrate the relationship between synthetic SFT dataset size and model performance on GSM8K and MATH. The curve clearly exhibits a scaling law relationship between the size of SFT data and model's performance. Here are some in-depth observations:

**Quantity Breeds Quality.** To enhance the mathematical reasoning abilities in LLMs, increasing the quantity of synthetic data can significantly improve the quality of model performance. This scaling trend implies that, while SFT with a small amount of data could achieve decent results [33], utilizing a larger scale of synthetic SFT data can further improve math reasoning performance.

**Diminishing Returns from Continual Pre-Training.** The DeepSeekMath-Base [23] 7B model, which has been continually pre-trained with 120B math-related tokens sourced from the web, initially demonstrates superior performance. However, as we increase the synthetic dataset size in the Skywork-MathQA dataset, this advantage diminishes and is eventually surpassed by the Mistral [15] 7B base model. As the amount of SFT data increases, Skywork-Math-Mistral-7B and Skywork-Math-LLaMA2-7B catch up in performance to the Skywork-Math-DeepSeekMath-7B. This suggests that while specialized pre-training provides a strong initial boost, its benefits are not consistently scalable and can be matched by increasing the quantity of synthetic SFT data.

**Effect of Problem Difficulty.** The accuracy performance for Skywork-Math 7B model series significantly increases as the synthetic data size expands from 2.1M to 2.5M, corresponding to the stage 2 in our data synthesis pipeline. This performance improvement in the final stage of data scaling indicates that incorporating more complex problems— ranging from Level 3 to Level 5 in the MATH dataset—has a substantial positive impact on model performance. This finding underscores the importance of not only generating a large quantity of data but also including more challenging problems to push the limits of math reasoning abilities of LLM models.

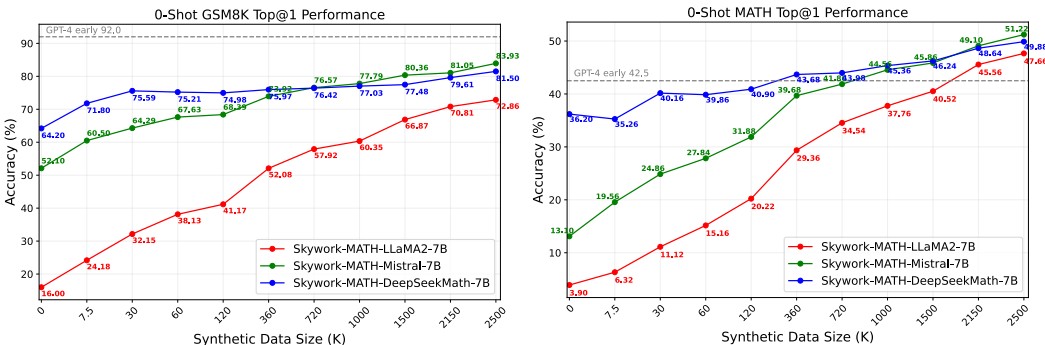

Figure 3: The zero-shot top1 performance of Skywork-Math 7B model series improves significantly with the increased size of synthetic SFT data in the Skywork-MathQA dataset, showing a clear trend of enhanced math as data quantity increases.

# 4 Conclusion

We study how to empower mathematical reasoning abilities for common 7B pre-trained LLM models. We propose the Skywork-MathQA dataset, consisting of 2.5 million diverse and high-quality SFT instances, implemented through our novel two-stage data synthesis pipeline. We introduce Skywork-Math model series, demonstrating that common small-scale 7B language models can stimulate strong mathematical reasoning ability using only synthetic SFT data. Skywork-Math models achieve state-of-the-art accuracy among models smaller than 10B parameters using only synthetic SFT data, surpassing 70B LLM models and an early version of GPT-4 on MATH. These results suggest that the data scaling law for mathematical reasoning in LLM models remains significant and promising. Notably, this research provides several valuable insights and practical takeaways to advance our understanding of the capabilities and limitations of LLMs in mathematical reasoning.

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
