# OpenReview forum: "Skywork-Math: Data Scaling Laws for Mathematical Reasoning in LLMs — The Story Goes On"
_NeurIPS.cc/2024/Workshop/MATH-AI — MATH-AI 24_

### Official Review · Reviewer_gFfk · 2024-09-29

**Rating:** 7
**Confidence:** 3

**Review:**

## Paper Summary
The paper investigates how mathematical reasoning in LLMs can be improved by scaling the quantity of training data. It introduces the Skywork-Math model series, fine-tuned on a synthetic dataset, which demonstrates superior performance on mathematical benchmarks and surpasses early versions of GPT-4.
## Pros
* **Novel Data Scaling Approach:** Overall, the paper has a high novelty in its proposed method. It introduces the Skywork-Math model series, which adopts a two-stage SFT approach with data synthesis pipelines to effectively demonstrate how scaling SFT data can enhance mathematical reasoning in LLMs. This focus on data scaling laws offers valuable insights for future research and industry applications.

* **Scalable and Efficient Use of Small Models:** A highlight in the paper is that the Skywork-Math models, with only 7B parameters, achieve results on par with larger models (up to 70B parameters) by leveraging fine-tuning and data scaling.  The results demonstrate that the method can maintain high accuracy and is resource-efficient.

* **Impressive Results on Benchmarks:** The Skywork-Math models outperform several state-of-the-art models, including an early version of GPT-4, on the MATH benchmark. Achieving accuracies of 51.2% on the MATH benchmark and 83.9% on GSM8K using only SFT data is a strong testament to the effectiveness of the method.

## Cons
* **Additional Benchmarks for broaden applicability:** This paper can be further improved by testing the method on additional math benchmarks to evaluate the proposed method's reasoning ability such as logical reasoning.

---

### Official Review · Reviewer_JNwV · 2024-10-05
**The paper seems good, but the first 4 pages don't have enough information**

**Rating:** 6
**Confidence:** 5

**Review:**

The paper has a total of 35(!) pages with the first 4 having very little information except high level conclusions. I didn't read the full 35 pages, so my review is based mostly on the first 4 plus a couple of figures and tables I selectively chose from the appendix.

The only figure related to the presented highlights in the first 4 pages is Figure 3. I think it's an interesting and practically important observation that continued training does not necessarily have much impact after SFT. I wonder if the hyperparameters for those separate models have been tuned separately or if the hyperparameters (most importantly LR) for Mistral/Llama have been used for DeepSeek directly. This might invalidate conclusions as models after continued training might need very different HPs for best performance.

The only other ablation mentioned in the first 4 pages is on the problem difficulty level, which is an interesting result. Although the numbers presented in Table 2 (in the appendix) are a bit inconclusive if we look at all 3 models. E.g. Mistral-7b seems to get an impressive 4 points boost in level 5 problems, while llama and deepseek slightly decrease. Also to make a fair comparison that indeed hard problems are the key, it's important to compare not with the baseline of 2.1M samples, but with 2.5M where we add random or easy problems (since maybe it's just the extra 0.4M of data that lead to this accuracy increase). Maybe this comparison is presented somewhere in the appendix, but I didn't read all 35 pages.

Overall, I think it's a good paper, but the first 4 pages have so little information that it's hard to judge that for sure.

---

### Official Review · Reviewer_7nob · 2024-10-07
**Outstanding in all aspects; therefore, accepted.**

**Rating:** 7
**Confidence:** 4

**Review:**

The paper explores the potential of LLMs to solve mathematical problems from a data-centric perspective, finding that scaling laws remain applicable.

Strengths:
1. The article is well-structured, well-written, and easy to understand.
2. The volume of experiments is substantial, with thorough experimental analysis, exceeding typical workshop standards.

Weaknesses: Some conclusions, such as the discussion on data quality and the impact of data scale on LLMs solving mathematical problems, lack novelty.

---

### Official Review · Reviewer_qq73 · 2024-10-07
**Skywork-Math Review**

**Rating:** 7
**Confidence:** 3

**Review:**

Summary
* This work introduces a 2.5M instance Skywork-MathQA dataset, which is generated by a novel data pipeline for generating quality instances for SFT
* They show that models fine tuned using the synthetic data can achieve SOTA accuracy even on models smaller than 10B parameters and that SFT with high quality data helps models build reasoning capabilities

Strengths
* Skywork-Math models are able to achieve state-of-the-art accuracy among models smaller than 10B parameters on the GSM8K and MATH datasets
* Scaling law findings are valuable, especially that there are diminishing returns to pre-training and that varying problem difficulty is important

Weaknesses
* Since the description of methods is so brief, the methods and the main contributions in the methodology are not very clear from the main body of the paper. It would help to move some of the Appendix B content into the main body.
* Missing analysis of the limitations/tradeoffs of the proposed two-stage method. Is there a ceiling where increasing the quantity of synthetic SFT data stops helping?

---

### Decision · Program_Chairs · 2024-10-09

Accept